# Chemical Probes for the Adenosine Receptors

**DOI:** 10.3390/ph12040168

**Published:** 2019-11-12

**Authors:** Stephanie Federico, Lucia Lassiani, Giampiero Spalluto

**Affiliations:** Department of Chemical and Pharmaceutical Sciences, University of Trieste, Via Licio Giorgeri 1, 34127 Trieste, Italy; lassiani@units.it (L.L.); spalluto@units.it (G.S.)

**Keywords:** adenosine receptors, fluorescent ligands, radioligands, radiotracers, covalent ligands, GPCR probes

## Abstract

Research on the adenosine receptors has been supported by the continuous discovery of new chemical probes characterized by more and more affinity and selectivity for the single adenosine receptor subtypes (A_1_, A_2A_, A_2B_ and A_3_ adenosine receptors). Furthermore, the development of new techniques for the detection of G protein-coupled receptors (GPCR) requires new specific probes. In fact, if in the past radioligands were the most important GPCR probes for detection, compound screening and diagnostic purposes, nowadays, increasing importance is given to fluorescent and covalent ligands. In fact, advances in techniques such as fluorescence resonance energy transfer (FRET) and fluorescent polarization, as well as new applications in flow cytometry and different fluorescence-based microscopic techniques, are at the origin of the extensive research of new fluorescent ligands for these receptors. The resurgence of covalent ligands is due in part to a change in the common thinking in the medicinal chemistry community that a covalent drug is necessarily more toxic than a reversible one, and in part to the useful application of covalent ligands in GPCR structural biology. In this review, an updated collection of available chemical probes targeting adenosine receptors is reported.

## 1. Introduction

Since their discovery in the mid seventies [1,2], adenosine receptors (ARs) have attracted research interest for their implication in a wide range of physiological and pathological processes (i.e., asthma, ischemia, cancer, Parkinson’s disease, etc.) [3]. As a consequence, at the same time research commenced on specific receptor probes that are essential tools for receptor characterization [4,5]. ARs exist as four different subtypes: A_1_, A_2A_, A_2B_ and A_3_ ARs [6,7]. Due to the advancement in techniques for detection and characterization of receptors, and in particular of G protein-coupled receptors (GPCRs) [8,9,10,11,12], the availability of suitable probes is a constant need. In particular, this review covers three specific chemical probe families for ARs: radioactive, covalent and fluorescent ligands. Radioactive ligands, properly called radioligands, are the oldest class of AR probes, and still represent the principal tool in drug discovery since their use in binding assays [7]. Recently, the broad interest in radioactive ligands is due to their development as radiotracers in positron emission tomography (PET), leading to new diagnostic possibilities [13,14]. On the other hand, covalent ligands for GPCRs, which were in the past used as tools to purify, isolate or pharmacologically characterize receptors, have recently attracted the interest of the scientific community for their ability to stabilize their target protein, increasing the probability of obtaining X-ray crystal structures [10]. This strategy was successfully applied for the A_1_ AR subtype [15,16]. Since A_2B_ and A_3_ AR crystal structures are still lacking, it is easy to imagine that several works will focus on development of covalent ligands for these receptor subtypes in the near future. Finally, the last few years have been characterized by the application of a variety of fluorescence-based methods for GPCR structure biology and drug discovery [17]. These techniques involve the introduction of a fluorescent tag on a GPCR or on a GPCR ligand, leading to fluorescent ligands, which are discussed here [9,18,19,20]. The aim of this review is to give a panorama of the available chemical probes for the ARs to researchers working in this field or medicinal chemists working on ARs or other GPCR targets.

## 2. Radioligands and Radiotracers

It is well known, that radioligand probes are useful for studying both the distribution and functions of receptors. In this class of compounds, two families of derivatives should be considered: i) radioligands, generally tritiated or iodinated, for binding studies; ii) radioligands used for imaging, in general probes including isotopes such as ^11^C, ^18^F and ^15^O. 

In the first class of compounds, in the last decades, several examples of radioligands for all AR subtypes, both agonists and antagonists, with different degrees of potency and selectivity have been reported and extensively reviewed [7,21,22,23,24]. Our purpose is to give a brief update of the work developed in this field in this review.

Considering labeled derivatives for binding studies only an agonist for A_2B_ AR named [^3^H]-BAY60-6583 (**1**) was recently reported by the group of Prof. C.A. Müller (Figure 1) [25]. 

This partial agonist in its tritiated form (the position of tritium is not reported) failed to be a good probe for binding studies. This is probably due to its moderate affinity at the human A_2B_ receptor and high level of non-specific binding. The only results obtained using this radioligand indicate that nucleoside and non-nucleoside agonists most probably bind the receptor in different conformations [25]. 

In contrast to the development of tritiated or ^125^I radioligands, in the last few years great efforts have been made in the field of radiotracers for imaging [14]. In particular, several examples of ^11^C or ^18^F derivatives for the different AR subtypes have been reported.

Regarding the ^11^C derivatives, some recent examples (compounds **2**–**5**) have been reported in Figure 2, in particular, regarding the A_1_ and A_2A_ ARs. 

One of the most studied derivatives is xanthine derivative [1-methyl-^11^C]8-dicyclopropylmethyl-1-methyl-3-propylxanthine (^11^C-MPDX, **2**). This compound has been utilized for various studies [26,27,28]. For example, Paul and coworkers performed in vivo binding studies in rats hypothesizing that agonists and antagonists may bind in different sites at the A_1_ receptor [26]. The same derivative was also utilized to evaluate A_1_ AR alterations in patients with chronic diffuse axonal injury [27] or to investigate the density of A_1_ ARs in patients with early-stage Parkinson’s disease [28].

Another interesting ligand for the A_1_ AR, is the pyridine derivative **3** reported by Guo et al. This compound could be considered the first brain blood barrier (BBB)-permeable A_1_ partial agonist with promising properties of detection of endogenous adenosine fluctuations [29].

Also in the field of A_2A_ AR antagonists, some labeled compounds have been recently reported, in particular the styryl-xanthine derivative [7-methyl-^11^C]-(E)-8-(3,4,5-trimethoxystyryl)-1,3,7-trimethylxanthine (^11^C-TMSX, **4**) that has been utilized for demonstrating that A_2A_ receptors could be a future target to enhance brown adipose tissue (BAT) metabolism [30]. In addition, the non-xanthine labeled derivative ^11^C-preladenant (**5**) has been reported to be an interesting probe for investigating cerebral distribution of A_2A_ ARs and other kinetic aspects such as its extensive distribution in the striatum [31,32].

Nevertheless, one of the major problems of the use of ^11^C-labeled compounds is the short half-life of this isotope (20 min), this means that the compound has to be prepared and immediately utilized before its inactivation. For this reason, other isotopes are preferred to ^11^C such as ^18^F, which has a half-life of 109 min. In this field, some promising agents for A_1_, A_2B_ and A_3_ receptors (**6**–**10**) have been reported (Figure 3).

One of the most used ^18^F-labeled compounds could be considered to be the xanthine derivative ^18^F-8-cyclopentyl-3-(3-fluoropropyl)-1-propylxanthine (^18^F-CPFPX, **6**). This compound has been extensively used for developing suitable pharmacokinetic models for the quantification of cerebral A_1_ ARs [33].

It has also been utilized to visualize and quantify the in vivo occupancy of the human cerebral adenosine A_1_ receptor by caffeine [34], and demonstrated that its chronic use did not lead to persistent changes in functional availability of A_1_ ARs [35].

Taking into account the results obtained with these derivatives, Kreft and coworkers developed other ^18^F-CPFPX structurally-related derivatives named CBCPM (8-cyclobutyl-1-cyclopropymethyl-3-(3-fluoropropyl)xanthine, **7**) and CPMMCB (1-cyclopropylmethyl-3-(3-fluoropropyl)-8-(1-methylcyclobutyl)xanthine, **8**), respectively, which showed similar behavior to the reference compound in preliminary PET studies [36].

In addition, a promising labeled derivative, named ^18^F-FE@SUPPY (5-(2-fluoroethyl)2,4-diethyl-3-(ethylsulfanylcarbonyl)-6-phenylpyridine-5-carboxylate, **9**), has been developed for the A_3_ AR, which has been shown to be a tracer for studying pharmacokinetic aspects and mapping A_3_ ARs in rats [37,38].

Very recently, a ^18^F-labeled form of a pyrazine A_2B_ AR antagonists (**10**) has been reported by Lindemann et al. [39]. Preliminary studies on this compound, which showed binding affinity in the nanomolar range towards the human A_2B_ AR, clearly indicated the potential of this derivative as a tracer for studying this receptor subtype and provided an important basis for the development of new tracers for A_2B_ AR [39].

## 3. Covalent Ligands

It is well known that covalent ligands may be considered important pharmacological tools for the study of the receptors; in fact, the irreversible blockade of the receptors could reveal details of both structural binding interactions and the patho-physiological role of the examined receptor.

The strategy utilized for obtaining irreversible ligands is generally based on the introduction of chemoreactive moieties on the scaffold of well-known agonists and/or antagonists. The limiting step in preparing this kind of probe is the position on which the chemoreactive group should be introduced in order to obtain irreversible block of the receptor without losing both affinity and selectivity. For this reason, several structure–activity relationship studies (SAR) have been performed in order to optimize the structural requirements that are indispensable for obtaining irreversible probes that retain affinity and selectivity.

In the field of ARs, several examples of covalent agonists and antagonists have been reported in the last decades. We wish, for this reason, to briefly summarize the results obtained in this research area.

Regarding agonists for ARs, several examples of A_1_, A_2A_ and A_3_ receptor subtypes are reported and depicted in Figure 4 and Figure 5.

The strategy to obtain irreversible agonists for ARs is based on the core modification of the natural nucleoside adenosine.

Concerning the A_1_ AR subtype, examples of covalent agents are reported and summarized in Figure 4 [40,41,42].

In particular the *N*^6^ substituted derivative of adenosine, named m-DITC-ADAC (4-isothiocyanatophenylaminothiocarbonyl, DITC; adenosine amine congener or *N*^6^-[4-[[[4-[[[(2-Aminoethyl)amino]carbonyl]methyl]-anilino]carbonyl]methyl]phenyl]adenosine, ADAC) (**11**), proved to be a good irreversible agonist for A_1_ AR. It was able to mimic ischemic preconditioning in rabbits [40] and also prolong the stimulus of the His bundle (SA) interval by 2.1 fold in guinea pig isolated hearts [41].

Another interesting covalent agonist, R-AHPIA (2-azido-*N*-(2-(4-hydroxyphenyl)-1-methylethyl)-adenosine, **12**) was reported by Klotz and coworkers [42]. This ligand was also labeled with ^125^I and, after UV irradiation, it irreversibly binds to the rat receptor, which was isolated for the first time as a protein of about 35 KDa [42].

A quite extensive SAR study on the adenosine nucleus for obtaining other covalent ligands for human ARs was performed by Jorg and coworkers [16], who obtained a series of compounds of general formula **13**. Nevertheless no extensive pharmacological studies for their characterization were reported [16].

Development of covalent A_2A_ agonists was performed by an extensive SAR study on the CGS21680 (2-*p*-(2-carboxyethyl)phenethylamino-5′-*N*-ethylcarboxamidoadenosine) scaffold, by introduction of several chemoreactive groups on its carboxylic acid moiety (Figure 5) [24,43,44]. 

These studies led to the discovery of a compound named DITC-APEC (2-[(2-aminoethyl-aminocarbonylethyl)phenylethylamino]-5’-*N*-ethyl-carboxamidoadenosine, APEC) (**14**), which was able to induce concentration dependent coronary vasodilation that persisted for a long time, even after washout. This suggests that compound **14** irreversibly bound A_2A_ AR in guinea pig coronary arteries [44].

A quite similar compound (**15**) was obtained by introducing a reactive ester on the CGS21680 nucleus as an acylating agent. This derivative was found to be an irreversible agonist for the human A_2A_ AR, and docking studies suggested that two lysine residues in the second extracellular loop were involved in the covalent binding. In fact, site directed mutagenesis studies confirmed that lysine 153 reacts with the active ester moiety, giving the irreversible adduct [45].

For the rat A_3_ AR subtype, an irreversible agonist (**16**) was developed by introducing a benzyl-isothiocyanate group at the *N*^6^ position of 5’-(*N*-methylcarboxamido)adenosine, but, except for binding experiments confirming the irreversible properties of the compound, no significative pharmacological experiments were performed [46].

In addition, several irreversible antagonists for ARs have been reported. We could classify the obtained compounds in two major classes: i) xanthine derivatives (the natural antagonists for ARs), and ii) non-xanthine derivatives.

In the xanthine group, several derivatives of the well-known A_1_ AR antagonist DPCPX (1, 3-dipropyl-8-cyclopentyl xanthine) have been synthesized by introducing chemoreactive groups [47,48,49]. Among this large number of synthesized compounds, the most interesting derivative could be considered to be 8-cyclopentyl-3-*N*-[3-((3-(4-fluorosulphonyl)benzoyl)-oxy)-propyl]-1-*N*-propyl-xanthine (FSCPX, **17**), which proved to be the best irreversible rat A_1_ AR antagonist reported [47] (Figure 6).

Several pharmacological studies have been performed using FSCPX (**17**) as a probe, permitting better understanding of the pathophysiological role of the A_1_ AR [50,51].

A structurally related derivative to FSCPX (**17**), bearing a cyclohexyl group at the 8 position and an amido function instead of an ester at the 3 position, named DU172 (**18**), has been synthesized and utilized to obtain a co-crystalized structure of the human A_1_ AR. With the help of computational and mutational studies, the structural basis for subtype selectivity versus A_2A_ subtype were identified [15].

A quite similar approach for developing other A_1_ adenosine covalent antagonists has been utilized using XAC, (*N*-(2-aminoethyl)-2-[4-(2,3,6,7-tetrahydro-2,6-dioxo-1,3-dipropyl-1*H*-purin-8-yl)phenoxy]-acetamide), as a template. This approach led to the discovery of several promising covalent agents such as m-DITC-XAC (**19**), p-DITC-XAC (**20**) [52], and the trifunctionalized compound **21** [53], which were found to be irreversible ligands for rat A_1_ ARs in binding studies. In particular, compound **21** was found to be 894-fold selective for A_1_ vs. A_2A_ ARs (Figure 6) [53]. 

By modifying the xanthine nucleus, other chemical entities have been developed by Scammells and coworkers. These studies led to some quite interesting covalent antagonists towards hamster receptors but low pharmacological characterization was reported [54].

In this field, some styrylxanthines have been investigated with the aim to obtain novel irreversible A_2A_ AR antagonists. These studies led to the discovery of ISC (8-(3-isothiocyanatostyryl)caffeine, **22**), which in binding studies was found to be a good covalent ligand at the rat receptor, with potency in the nanomolar range and good levels of selectivity (Figure 6) [55].

Also in the non-xanthine family, several examples of irreversible AR antagonists have been reported [56,57,58,59].

In order to obtain covalent A_2A_ antagonists, insertion of a fluorosulfonyl moiety on the well-known A_2A_ AR antagonists SCH58261 (2-(2-Furanyl)-7-(2-phenylethyl)-7*H*-pyrazolo [4,3-*e*][1,2,4]triazolo[1,5-*c*]pyrimidin-5-amine) and ZM241385 (4-(2-(7-amino-2-(furan-2-yl)-[1,2,4]triazolo[1,5-*a*][1,3,5]triazin-5-ylamino)ethyl)phenol) have been developed. This approach led to the discovery of FSPTP (5-amino-7-[2-(4-fluorosulfonyl)phenylethyl]-2-(2-furyl)-pryazolo[4,3-*e*]-1,2,4-triazolo[1,5-*c*]pyrimidine, **23**) and LUF7445 ((4-((3-((7-amino-2-(furan-2-yl)-[1,2,4]triazolo[1,5-*a*][1,3,5]triazin-5-yl)amino)propyl)carbamoyl)benzene sulfonyl fluoride), **24**) as irreversible A_2A_ AR antagonists (Figure 7). 

The use of FSPTP (**23**) in pharmacological studies showed that there is a large receptor reserve for the A_2A_ AR, which mediates the increase in coronary conductance in guinea pigs [56], while LUF7445 (**24**), through the help of computational and mutational studies permitted identification of lysine 153 as the amino acid involved in covalent binding at the hA_2A_ AR [58].

A quite similar approach has been utilized for developing covalent human A_3_ AR antagonists, leading to a derivative named LUF7602 (4-((3(8-methoxy-2,4-dioxo-3-propyl-3,4-dihydropyrido[2,1-*f*]purin-1(2*H*)yl)propyl)carbamoyl)benzenesulfonyl fluoride, **25**) [57] and a pyrazolo-triazolo-pyrimidine compound (**26**) [59]. The first one showed that tyrosine 265 was involved in the covalent binding, while compound **26**, through the help of computational studies, suggested that serine 247 or cysteine 251, both in TM6, could be responsible for covalent binding. Of course, these results clearly indicate that the two compounds have a different binding pose into the receptor binding site.

## 4. Fluorescent Ligands 

The concept of using fluorescent ligands to detect GPCRs in native conditions, including ARs, is not new [60]. However, the urgent need for new fluorescent tools for ARs is given by the possibility of applying new fluorescent techniques to deeply study the receptors in cells, in particular, receptor localization, both on the membrane surface and during their internalization. This would allow for study of desensitization, recycling and homo- or oligo-merization of the receptors [19]. A description of the techniques used to study GPCRs was recently reviewed and comprised fluorescence polarization, confocal microscopy, fluorescence correlation spectroscopy (FCS), flow cytometry, FRET and bioluminescence resonance energy transfer (BRET) [18,19,61,62,63]. Nowadays, thanks to the progress in both synthetic and instrumental techniques, a plethora of fluorophores is available, with a diversity of applicable conjugation strategies that simplify the development of new specific fluorescent ligands [64,65,66]. In fact, the literature reports several examples of fluorescent ligands for all four AR subtypes. A very comprehensive review on fluorescent ligands for ARs was reported by Kozma et al. [67] and extended to some newer derivatives by another two reviews published in 2014 and 2015 [68,69]. Thus, here the most representative fluorescent ligands are reported among with the new fluorescent ligands for the ARs reported in the literature from 2015 to now.

The first fluorescent ligands were developed in 1987 by Jacobson and coworkers to target the rat A_1_ AR. The authors conjugated the fluorophores fluorescein isothiocyanate (FITC) and nitrobenzoxadiazole (NBD) to the free amino group of the A_1_ agonist ADAC (FITC-ADAC, **27**; NBD-ADAC, **28**), maintaining affinity to the receptor (pKi@ratA_1_ = 8.14, **27**; pKi@ratA_1_ = 8.36, **28**). The same approach was used for conjugating FITC to the A_1_ antagonist XAC, FITC-XAC (**29**, pKi@ratA_1_ = 6.90) (Figure 8) [60].

Therefore, fluorescent ligands for human adenosine receptors were developed both from agonists and antagonists. Concerning agonists, only few derivatizations were made from the simple adenosine scaffold. Specifically, in ABA-X-BY630 (**30**) [70], adenosine was conjugated by a linker at the 6 position, while in compound **31** [71], thioadenosine was used to introduce the BODIPY (boron dipyrromethene) fluorophore at the 2 position of the purine ring (Figure 9). ABA-X-BY630 (**30**) was developed as an A_1_ ligand (pKi = 6.65), but compound **31** exhibited a preferential affinity towards the A_3_ AR subtype (pK_i_ = 6.21), showing more than ten-fold selectivity against A_1_ and A_2A_ ARs. More efforts were made with the more active, but still not selective, agonist NECA (5’-(*N*-ethylcarboxamido)adenosine). It has been studied by different groups, where all substitutions were carried out at the 6 position and linkers of different length were used, as were different fluorophores such as dansyl, NBD, BODIPY 630/650, Cy5, Texas Red, and EvoBlue30 [70,72,73,74,75]. In most cases, the introduction of a fluorophore led to A_1_ AR ligands. Macchia et al. developed a series of dansyl-NECA derivatives that exhibited higher affinity at the A_1_ AR (i.e., ABEA-dansyl, **32**) [75]. Because dansyl excitation wavelength (340 nm) falls in the autofluorescence spectrum of cells and tissues, the authors substituted the dansyl group with a longer excitation wavelength fluorophore such as NBD (λex = 465 nm, λem = 535 nm). Interestingly, NBD led to a detrimental effect on A_1_ AR, while the affinity towards A_3_ AR (i.e., NBD-NECA, **33**) increased [73] (Figure 9).

In contrast, Middleton et al. developed various NECA-BODIPY 630/650 derivatives as A_1_ ligands, because red-emitting BODIPY fluorophores become brighter in non-aqueous environments, therefore specific fluorescence on the plasma membrane surface can be more easily visualized. Among them, ABEA-X-BY630 (**34**) showed high affinity for both A_1_ and A_3_ ARs and maintained agonist behavior [70] (Figure 9). This allowed visualization of the receptors at the cell membrane by confocal microscopy and quantification of its binding kinetics. Consequently, it was used as a probe to investigate allosteric interactions given by small molecules and receptor dimerization [76,77]. Unfortunately, ABEA-X-BY630 (**34**) showed high levels of non-specific cytoplasmic uptake, which prevented its use in long-term studies. In a work aimed to improve these poor imaging properties, an additional tripeptide linker, A-A-G, was introduced between the fluorophore and the agonist to obtain ABEA-A-A-G-X-BY630. All four possible combinations of L and D amino acids were examined, and apart from the (L)A-(L)A-G isomer, all others showed excellent affinity values at the A_1_ and A_3_ AR. The ABEA-(D)A-(D)A-G-BY630 (**35**) agonist was selected for further experiments and because it showed activation of the Gs protein at the A_1_ AR when tested at high concentrations, this compound was preferentially used as a probe for the A_3_ AR (Figure 9). As an example, compound **35** was successfully used to investigate the internalization process of A_3_ ARs and their co-localization with arrestin [78].

As shown in Figure 10, among agonists, the only other compound that was conjugated to a fluorophore in order to obtain, in this case, an agonist for the A_2A_ AR subtype was APEC (2-[2-[4-[2-(2-aminoethylcarbonyl)ethyl]phenyl]ethylamino]-5′-*N*-ethylcarboxamidoadenosine). It was successfully functionalized with FITC, Alexa Fluor 488 (MRS5206, **36**) and Alexa Fluor 532 (MRS5424, **37**) [79,80,81]. MRS5206 (**36**) showed affinity towards both A_2A_ (K_i_ = 149 nM) and A_3_ (K_i_ = 240 nM) ARs and was used to demonstrate that A_2A_ agonist-induced internalization is mediated by a clathrin-dependent mechanism [80]. Instead, the APEC–Alexa Fluor 532 conjugate MRS5424 (**37**) was developed to be applied in FRET experiments. In fact, the A_2A_ receptor tagged with CFP (cyan fluorescent protein) was able to emit at 480 nm when excited at about 430 nm (blue). Then, emission at 480 nm was able to excite Alexa Fluor 532 in MRS5424 (**37**), which consequently emitted at 554 nm (yellow). In particular, MRS5424 (**37**) was used for the investigation of allosterism mediated by A_2A_/D_2_ oligomerization [8,81].

APEC was also used to develop quantum dot (QD) conjugates, which are well-known for their emission properties. In particular, a QD was functionalized, through a linker based on (R)-thioctic acid, with a dendron containing multiple molecules of APEC. Dendron was used to increase the solubility of the structure and to ensure a good loading of APEC on the QD. One of the prepared systems, MRS5303, showed affinity towards A_2A_ AR, but further work is needed to obtain a good A_2A_ probe [82]. 

Finally, Jacobson and coworkers reported different methanocarba derivative-based fluorescent probes for the A_3_ AR subtype. The functionalized scaffold is that of the A_3_ agonist MRS3558 ((1S,2R,3S,4R,5S)-4-[2-chloro-6-[(3-chlorophenyl)methylamino]purin-9-yl]-2,3-dihydroxy-*N*-methylbicyclo[3.1.0]hexane-1-carboxamide) in which the chlorine atom at the 2 position of the purine ring was substituted with an alkyne moiety, and where different fluorophores (e.g., Cy5, squaraine-rotaxane, Alexa Fluor 488, pyrene) were attached directly or through a linker [83,84,85]. Compound MRS5218 (**38**) is a full agonist of the A_3_ AR, displaying an IC_50_ of 1.09 nM in a functional cAMP assay (Figure 10) [83].

Concerning antagonists, the most studied fluorescent probes were based on the xanthine derivative XAC. As for NECA, several linkers and different fluorophores were introduced on the free amino group in XAC in this case, leading to both A_1_ and A_3_ AR fluorescent antagonists [60,72,86,87,88,89]. Among them XAC-X-BY630 (**39**) and CA200645 (**40**) are the more representative (Figure 11) [87,90]. 

The first aim was to develop a probe for the A_1_ AR, and from SAR analysis, it is known that the introduction of long chains at the 8 position of the xanthine ring are well tolerated by the A_1_ AR. In fact, XAC-X-BY630 (**39**) showed affinity in the nanomolar range towards A_1_ AR and it was used to quantify ligand-receptor binding at a single cell level using FCS [87]. Instead, CA200645 (**40**), which bears a longer polyamido linker even if it is particularly used as an A_3_ florescent probe (i.e., for kinetics, allosterism and dimerization studies by FCS and nano-BRET technology [91,92,93]), binds potently both to A_1_ and A_3_ ARs, in fact, it was used in a high-content screening assay for both receptors [90].

Further optimization on the linker of XAC/BODIPY 630 derivatives was reported by Vernall et al., where, like NECA derivatives, the authors decided to introduce a peptide linker, instead of a naked polyamido chain like in CA200645 (**40**) [89]. The aim was to increase selectivity at the A_3_ AR by gaining specific interactions with the linker. In order to find a less expensive way to assess the SAR of this compound, the BODIPY 630 was replaced by the similarly bulky Fmoc group. The linker was made by two amino acids chosen between alanine, tyrosine, serine and asparagine. The last were chosen for their polarity and, thus, ability to make hydrogen bonds. In addition, tyrosine can also be engaged in π-π interactions. The best results were obtained with XAC-(S)-S-(S)-Y-BY630 (**41**), which showed a pK_D_ of 9.12 towards the A_3_ AR, compared to a pK_D_ of 7.26 towards the A_1_ AR [89]. Compound **41** demonstrated a long residence time at A_3_ AR (RT = 288 min); thus, when it was used to determine the association and dissociation constants of the ligands, it gave quite different constants with respect to those determined with classical radioligands, such as [^3^H]PSB-11 ([^3^H]-(8R)-8-ethyl-1,4,7,8-tetrahydro-4-5*H*-imidazo[2,1-*i*]purin-5-one) [92]. This result suggested again that a probe should be properly selected before its use in an experiment.

XAC was also used to prepare GPCR ligand–dendrimer (GLiDe) conjugates. In particular, XAC was functionalized at the free amino group present at the 8 position, with an alkyne-containing chain. Alkyne was then used to perform a copper catalyzed click reaction with an azido-containing G4 (fourth-generation) polyamidoamine (PAMAM) dendrimer to form triazoles. Dendrimers were also conjugated to fluorophores such as Alexa Fluor 488, leading to fluorescent dendrimers MRS5397 and MRS5399. In addition, near infrared dyes were also used by Jacobson and coworkers, leading to MRS5421 (which used NIR dye 800) and MRS5422 (which used NIR dye 700). These compounds showed higher affinity towards the A_2A_ AR, followed by A_3_ and A_1_ ARs [88]. 

A series of xanthine derivatives was recently developed to obtain fluorescent antagonists towards the A_2B_ AR [94]. The xanthine scaffold was chosen on the basis of known SAR for this receptor, which suggested that 3-unsubstituted 8-phenyl-1-propylxanthine derivatives are generally more potent and selective to A_2B_ AR than the corresponding 1,3-substituted derivatives. Thus, the BODIPY fluorophore was conjugated to the 8-phenyl by means of spacers of different lengths. The best compound of the series was PSB-12105 (**42**) (Figure 12), which showed nanomolar or subnanomolar affinity towards human, rat and mice receptors and was consequently used to develop a binding assay using flow cytometry [94].

In addition, scaffolds other than xanthines have been used to develop fluorescent antagonists for the adenosine receptors. In particular, tricyclic antagonists that exhibited high affinity and selectivity for the various AR subtypes were used; among them we can find [1,2,4]triazolo[1,5-*c*]quinazolines [95], [1,2,4]-triazolo[4,3-*a*]quinoxalin-1-ones [96], pyrazolo[4,3-*e*][1,2,4]triazolo[1,5-*c*]pyrimidines [95,97,98,99,100] and benzo[4,5]imidazo[2,1-*c*][1,2,4]triazines [101].

CGS15943 (9-chloro-2-(2-furanyl)-[1,2,4]triazolo[1,5-*c*]quinazolin-5-amine) is a well-known non-selective AR antagonist that, when conjugated to Alexa Fluor 488 by a triazole-containing linker at the 5 position, led to MRS5449 (**43**, Figure 13) displaying a radioligand binding Ki value of 6.4 nM at the A_3_ AR, and was more than 10-fold selective against A_1_ and A_2A_ ARs. For these characteristics it was chosen for the establishment of an A_3_ AR biding assay using flow cytometry [95].

In contrast, the analog pyrazolo[4,3-*e*][1,2,4]triazolo[1,5-*c*]pyrimidine derivative SCH442416 (5-Amino-7-[3-(4-methoxy)phenylpropyl]-2-(2-furyl)-pyrazolo[4,3-*e*]-1,2,4-triazolo[1,5-*c*]pyrimidine), was used to design potent A_2A_ AR fluorescent antagonists. Linkers were introduced on the hydroxy group on the phenylalkyl chain present at the 7 position of the scaffold. Three different fluorophores were introduced: Alexa Fluor 488, Tamra and BODIPY 650/665, leading to MRS5346 (**44**), MRS5347 (**45**) and MRS5418 (**46**), respectively (Figure 13). MRS5418 (**46**) showed the best affinity (K_i_ = 15.1 nM) and selectivity for the A_2A_ AR [97,98], but MRS5346 (**44**) was used in a fluorescence polarization-based receptor binding assay at the A_2A_ AR [99]. Ciruela and coworkers developed another member of this class of A_2A_ antagonists, where they introduced dye 467 directly to the hydroxyl group of SCH442416 (Figure 13). SCH442416Dy647 (**47**) was employed to undoubtedly demonstrate the presence of A_2A_/D_2_ oligomers in native tissues of rat striatum, which is essential information for designing new therapies for Parkinson’s disease [69,100].

The pyrazolo[4,3-*e*][1,2,4]triazolo[1,5-*c*]pyrimidine scaffold was also functionalized at the 5 position, leaving the 7 position unsubstituted and inserting a methyl group at the 8 position: substitutions that should shift selectivity towards the A_3_ AR [102]. FITC was used as fluorophore and simple diamino alkyl or PEG linkers were used as spacers. These derivatives exhibited equal low micromolar affinity for the A_2A_ and A_3_ AR subtypes [99]. In an attempt to obtain more potent derivatives, an aromatic ring, which could gain some interactions with the receptor, was introduced. As expected, new compounds (**48** and **49**) displayed nanomolar affinity at the two receptors (Ki A_2A_ = 60.4 nM, Ki A_3_ = 73.6 nM, **48**; Ki A_2A_ = 90 nM, Ki A_3_ = 31.8 nM, **49**) (Figure 13) [103].

Vernall et al. developed a series of A_3_ fluorescent derivatives of the 4-amino-2-(4-methoxyphenyl)-[1,2,4]triazolo[4,3-*a*]quinoxalin-1(2*H*)-one, by introducing different fluorophores through a polyamido linker at the 4 position of the scaffold [96]. The presence of the *p*-methoxy moiety on the phenyl group at the 2 position of the triazoloquinoxaline ring and the introduction of an acyl on the amino group at the 4 position, are the characteristics that should confer affinity at the A_3_ AR. A mini-series of compounds was developed using different linkers (chains from 2 to 12 atoms containing C, N and O atoms) and fluorophores (e.g., BODIPY, Tamra and Cy5). The best compound of the series was AV039 (**50**), which showed a pK_D_ of about 9 at the human A_3_ AR and was more than 650-fold selective over other receptor subtypes (Figure 14). This compound was able to specifically label the A_3_ AR in cells containing a mixed population of ARs, thus it represented an ideal tool for the study of these receptors [92,96].

Recently, Barresi et al. reported a series of benzo[4,5]imidazo[2,1-*c*][1,2,4]triazines derivatives as fluorescent probes for the A_1_ and A_2B_ ARs. The authors used NBD as a fluorophore, which was introduced using two linkers of different lengths at the *N*^1^ or *N*^10^ positions, in order to investigate which position gives the best flexibility to the molecule, to best accommodate the orthosteric binding site [101]. One compound (**51**) (Figure 14) showed an affinity of about 1 µM at the A_1_ AR and an IC_50_ value in the same range towards the A_2B_ AR, and therefore was demonstrated to be a useful tool for detecting these AR subtypes in fluorescence confocal microscopy experiments on cells. 

## 5. Conclusions

From this review, it is clear that even if ARs were discovered more than forty years ago, their complex mechanism of action, which includes, but is not limited to, activation, oligomerization and internalization processes, is not fully elucidated. Over the last years, great progress in the field has been obtained thanks to the advancements made in structural biology. AR probes were continuously applied to new available techniques, which allowed deep investigation of the receptors, but at the same time, obtained information that is useful for designing new, more specific, drugs and probes, which in turn has contributed to the improvement in the AR research field. In particular, a better understanding of the role and the mechanism of action of these receptors could allow design of more specific therapies for several pathological conditions in which these receptors are involved, and for that reason, ARs remain a very interesting and intriguing therapeutic target.

## Figures and Tables

**Figure 1 pharmaceuticals-12-00168-f001:**
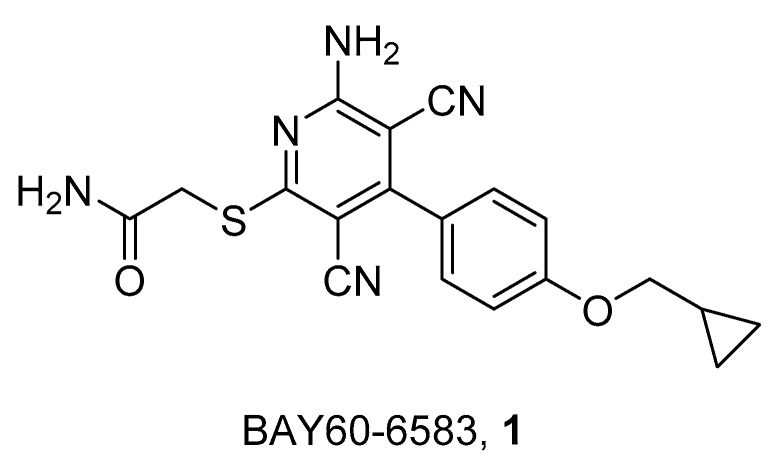
Structure of BAY60-6583.

**Figure 2 pharmaceuticals-12-00168-f002:**
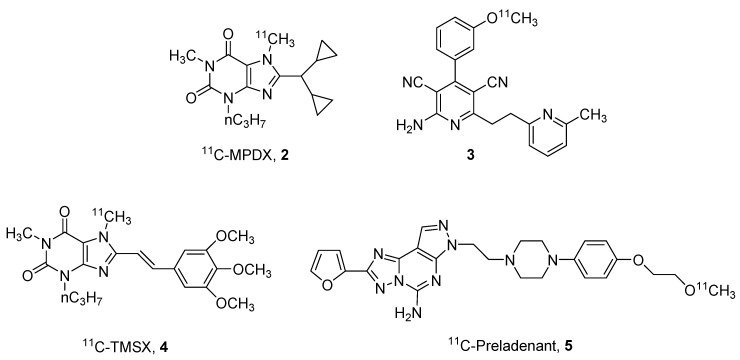
Structures of ^11^C labeled adenosine receptor (AR) ligands.

**Figure 3 pharmaceuticals-12-00168-f003:**
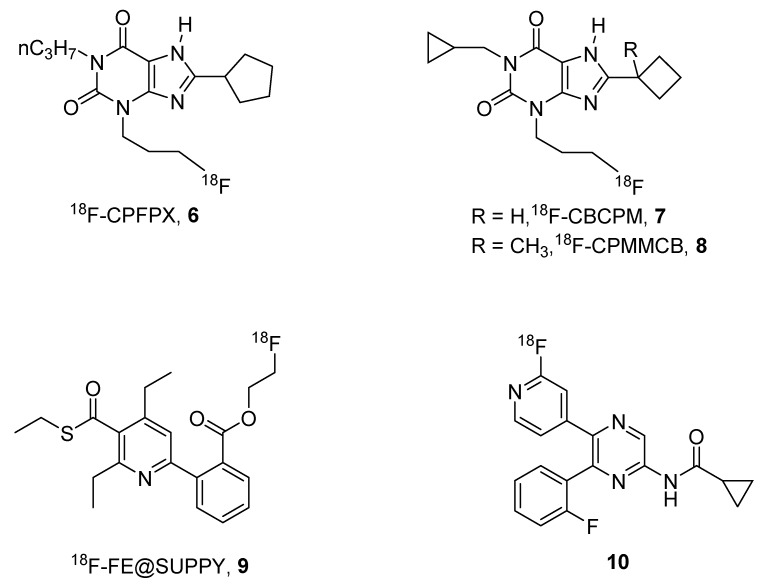
Structures of ^18^F-labeled AR ligands.

**Figure 4 pharmaceuticals-12-00168-f004:**
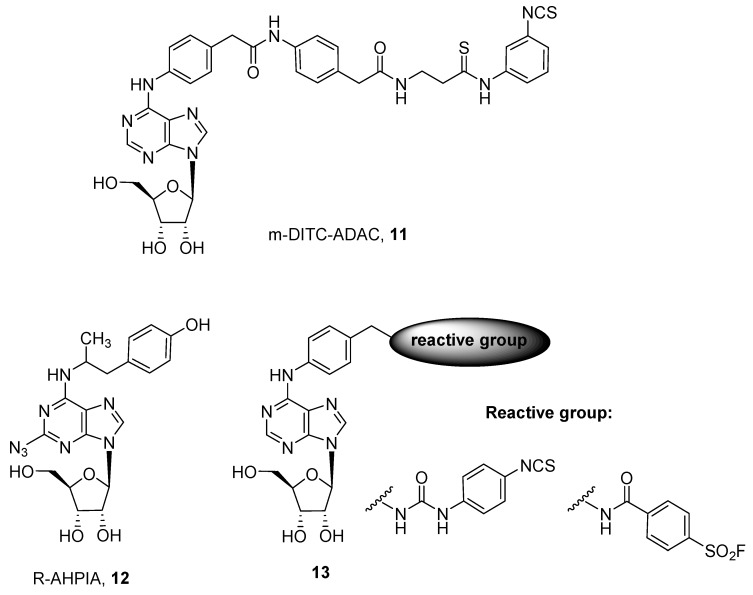
Structures of covalent A_1_ AR agonists.

**Figure 5 pharmaceuticals-12-00168-f005:**
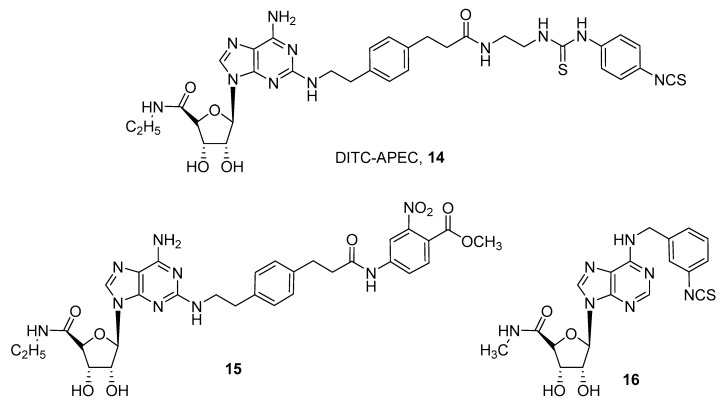
Structures of covalent A_2A_ and A_3_ AR agonists.

**Figure 6 pharmaceuticals-12-00168-f006:**
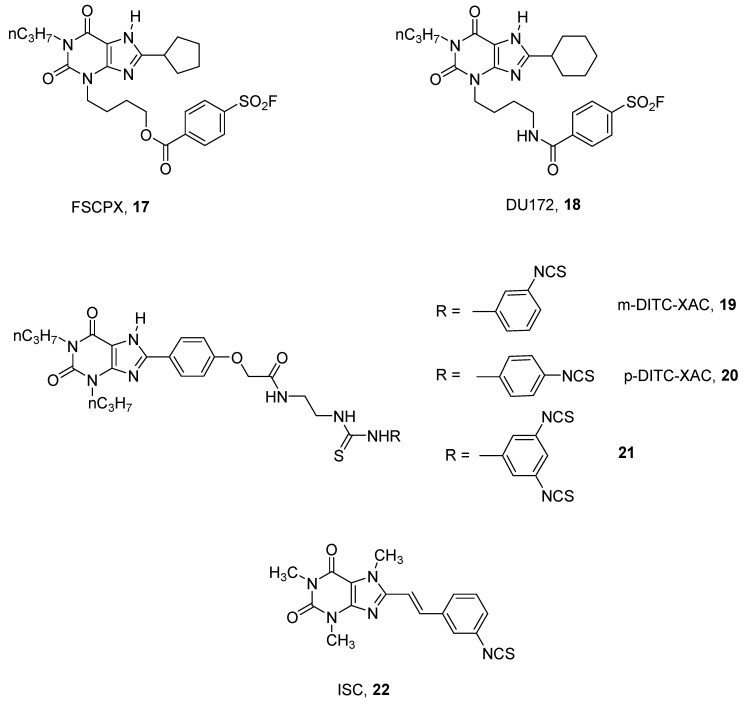
Structures of xanthine derivatives as covalent antagonists for ARs.

**Figure 7 pharmaceuticals-12-00168-f007:**
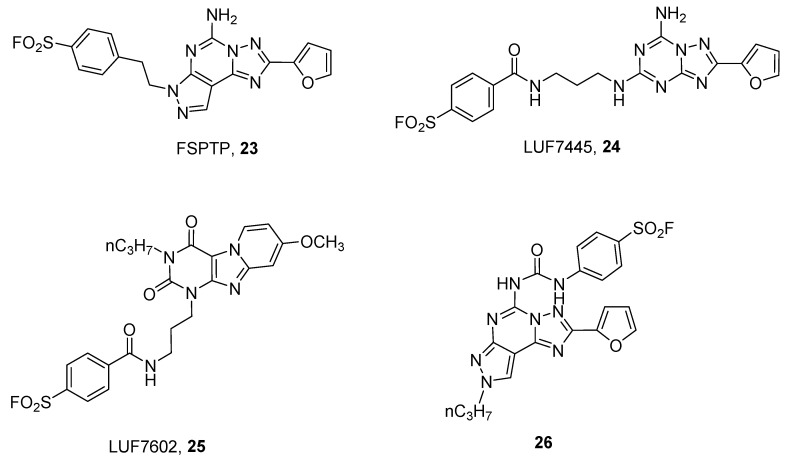
Structures of non-xanthinic derivatives as covalent antagonists for ARs.

**Figure 8 pharmaceuticals-12-00168-f008:**
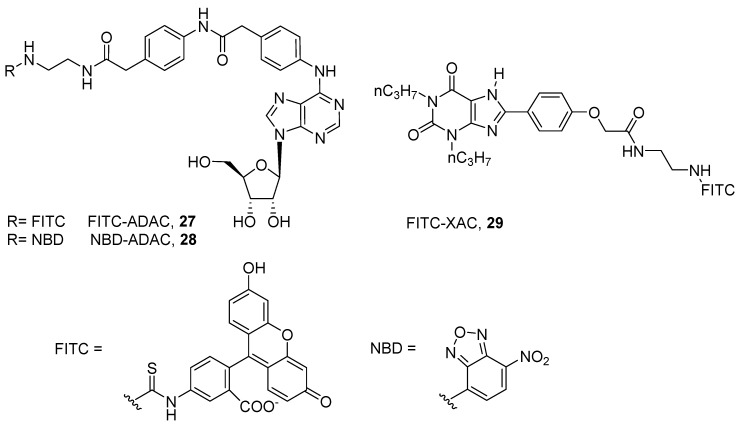
Structures of the first examples of fluorescent ligands for ARs.

**Figure 9 pharmaceuticals-12-00168-f009:**
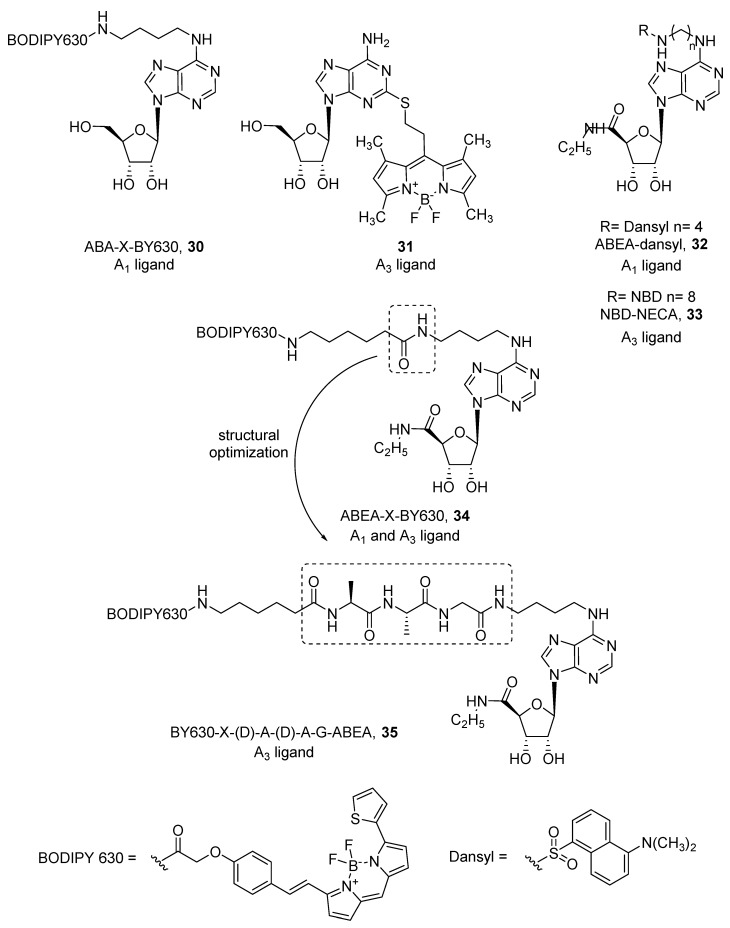
Structures of representative fluorescent agonist ligands for A_1_ and A_3_ ARs.

**Figure 10 pharmaceuticals-12-00168-f010:**
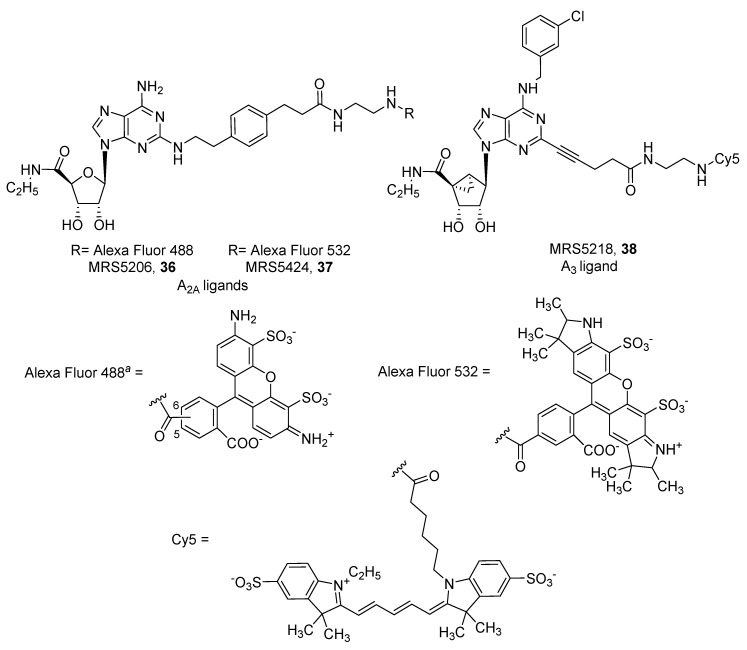
Structures of representative fluorescent agonist ligands for A_2A_ and A_3_ ARs.

**Figure 11 pharmaceuticals-12-00168-f011:**
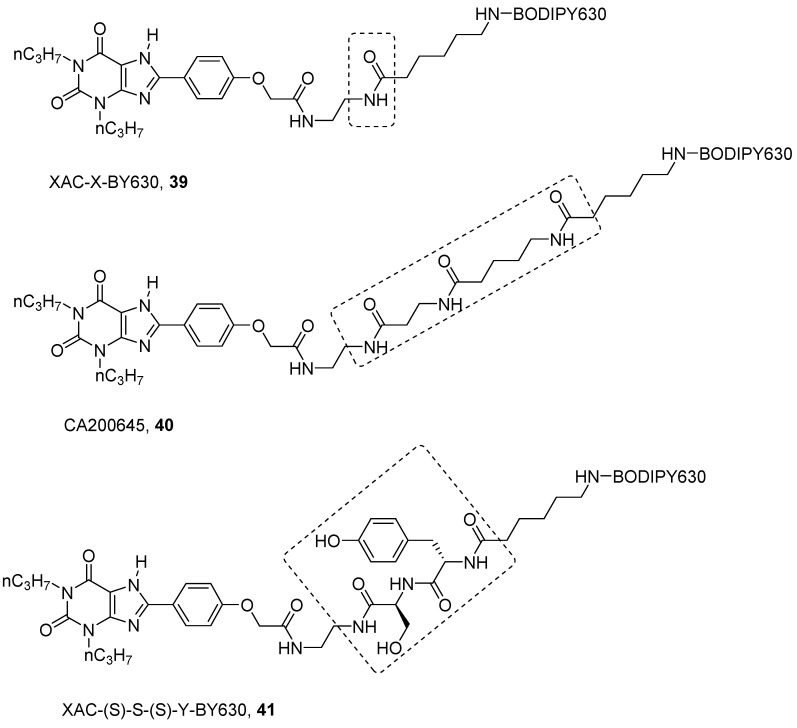
Structures of representative fluorescent xanthine antagonist ligands for A_1_ and A_3_ ARs.

**Figure 12 pharmaceuticals-12-00168-f012:**
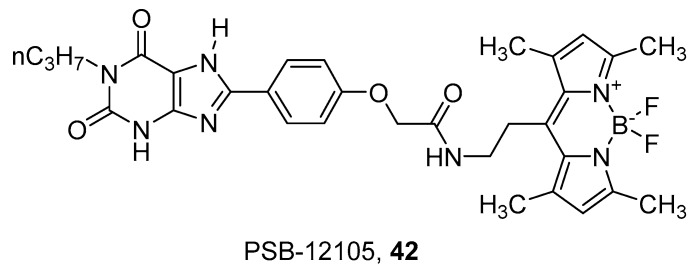
Structures of the fluorescent xanthine A_2B_ AR antagonist PSB-12105 (**42**).

**Figure 13 pharmaceuticals-12-00168-f013:**
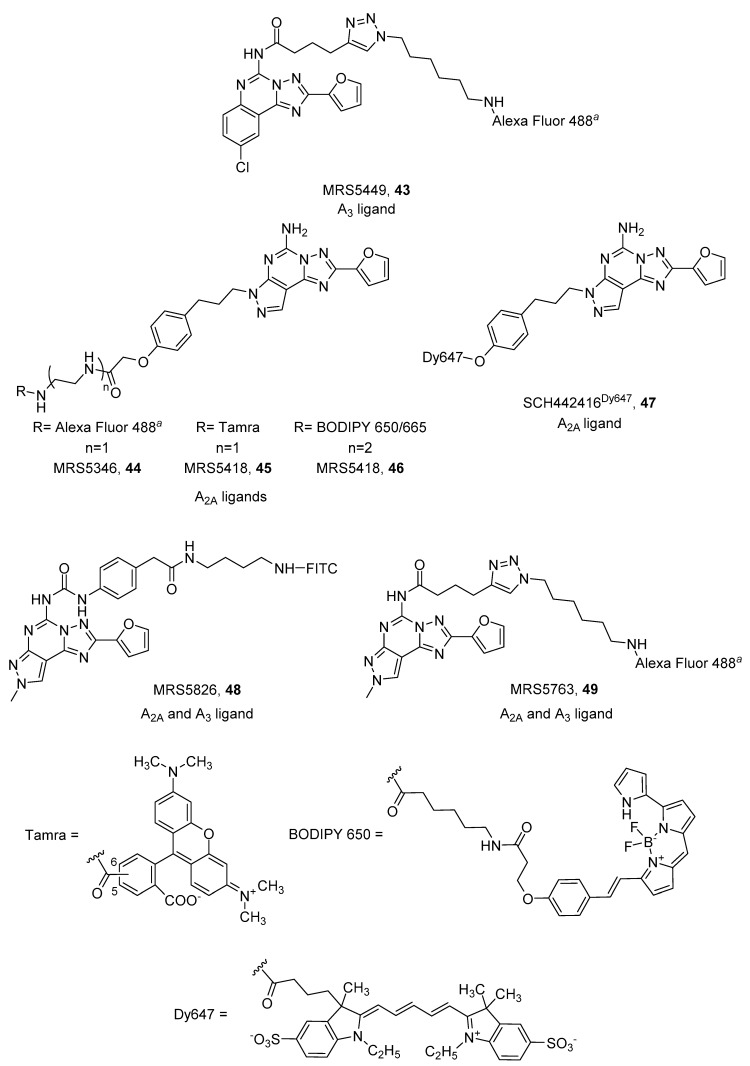
Structures of tricyclic fluorescent A_2A_ and A_3_ ARs antagonists. ^a^Alexa Fluor 488 5 isomer.

**Figure 14 pharmaceuticals-12-00168-f014:**
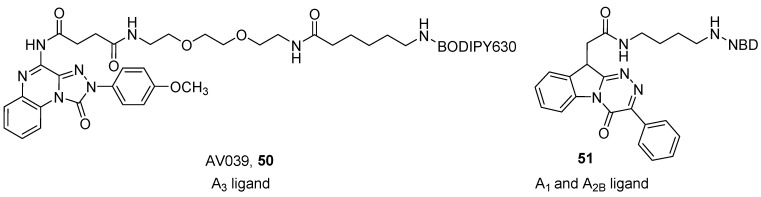
Structures of other tricyclic fluorescent AR antagonists.

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
