# Peer review of "Chemical Probes for the Adenosine Receptors"

_pharmaceuticals, 2019, doi:10.3390/ph12040168_

Round 1

Reviewer 1 Report

The review covers completely this important field of the research for the comprehension of the biological role of adenosine receptors, giving an interesting overview of the results obtained by various groups. The cited references are updated and cover the majority of the literature of the branch.

Anyway, I suggest to carefully control the manuscript for some grammar mistakes (i.e. line 90 mean instead means, line 101 lead instead leads and so on). Some Figure legends are not correct : Figure 6 Non-xanthinic? Figure 7 Covalent A1Ar?

The authors should check also the correct  structures reported: Figure 5 structure 16 (Methyl instead Ethyl). Figure 9 and 10 compound 32, 33, 34 and so on, NH group missing. 

Reviewer 2 Report

The present review by entitled “Chemical probes for the adenosine receptors” by Federico S. et al reports on new chemical entities tagged with radioactive atoms, or dyes and fluorophores or able to bind covalently to membrane protein and used to study adenosine receptors activity by newly available techniques.

This summary of the different molecules used as probes and the different approaches for their selection is useful for scientists working in the adenosine receptors research field and, therefore, I recommend the publication of the paper after major revision due to the following corrections that need to be undertaken.

Major corrections:

Check all the structure of designed molecules: figure 5, compound 16 is a methyl derivative while in the discussion (see text below figure) is described as a NECA derivative; figure 9, compounds 32-35 and figure 10 compounds 36-38 are not drawn as NECA derivatives; It would be useful to indicate at which receptor species, the affinity or the activity is referred: in particular, for the A3 subtype there is a big difference in affinity between human and murine receptors; Page 3: it is stated that in the article cited (26) it is demonstrated that agonists and antagonists bind in different receptor sites but in the article it is instead said that it is suggested (may be) different binding sites for agonists and antagonists. Furthermore, different crystal structures of A1 receptors are available in the PDB in which adenosine (agonist) and different antagonists bind in the same receptor cavity, refer to 6D9H (A1 AR crystallized with adenosine), 5N2S (A1 AR crystallized with the antagonist PSB36), 5UEN (A1 AR crystallized with covalent antagonist DU172). Hence, in my opinion, the statement "demonstrating" should be changed to "suggesting"; The text needs to be checked by a native speaker and some parts need to be rephrased; examples: lines 141, 271, 325: these phrases are not clear and should be rephrased; line 338: the phrase could be singular or plural and changed opportunely;

General corrections are:

when speaking of adenosine receptors, the code after the A should be subscript: example A2A, 2A should be subscript, and the same for A1, A2B, and A3; when speaking about N6 substituent N should be italic and 6 apices as N6 (line 136, line 162, line 277); the abbreviation can be used after it has been described in the full name; the references must always be placed in the same position and preferably, in this format, before punctuation (examples page 1, lines 29, 31) and before the references code space should be added (examples page 1, lines 34 and 36, etc.);

Minor corrections are:

In the abstract: “compounds’ screening” should be corrected to “compounds screening”, phrase in line 66: last few years used twice; line 69: (compounds 2-5), 2 should be bold; line 77: “agonist and antagonist” should be changed to “agonists and antagonists”; line 92: “promising agent” should be corrected to “promising agents”; line 100: “utilized for visualize” should be corrected to “utilized to visualize”; line 101: “receptor of caffeine” should be corrected to “receptor by caffeine”; line 107: “pharmacokinetics aspects” should be corrected to “pharmacokinetic aspects”; line 131: “natural the nucleoside” should be corrected to “the natural nucleoside”; line 138: “to prolonged” should be corrected to “to prolong”; line 143: “covalent ligand for ARs has been” should be corrected to “covalent ligands for ARs have been”; line 149: move “(Figure 5)” before the references; line 153: correct to “studies led to the discovery”; line 157: correct to “by introducing on the CGS21680 nucleus”; line 181: “aminoethyl” should be lower case; line 182: correct to “led to the discovery”; remove one space between agents such; line 185: move “(Figure 6)” before the references; line 188: move “other chemical entities” after nucleus; line 192: correct to “led to the discovery”; correct to “which, in binding studies,” line 194: move “(Figure 6)” before the references; line 202: move “(Figure 7)” before the references; line 243: remove one space before in ABA-X-BY630; line 268: “and additional tripeptide linker” should be corrected to “an additional tripeptide linker”; line 269: “agonist, ABEA-A-A-G-X-BY630” should be corrected to “agonist to obtain ABEA-A-A-G-X-BY630”; line 276: “this time” should be changed to “in this case”; line 278: “FITC” is not explained in the text; line 311: “FCS” is not explained in the text; line 312: correct to “used as an A3”; figure 13: there is a bond out of any structure; line 398: move “(Figure 14)” close to the number 51 otherwise it seems the figure refers to the assay; figure 14: correct in the structure of 51 “BDN” to “NBD”.

Round 2

Reviewer 2 Report

Dear authors,

in the revised version, I found the paper suitable for publication: all the questions were clarified and mistakes corrected.